# Toward Improving Working Conditions to Enhance Professionalism of Convention Workers: Focusing on the Difference between Job Satisfaction and Job Performance According to Professionalism Perception

**DOI:** 10.3390/ijerph19105829

**Published:** 2022-05-10

**Authors:** Wenyan Yan, Eunjin Kim, Soyeon Jeong, Yeonghye Yoon

**Affiliations:** 1Business Administration Major, Honam University, Gwangju 62399, Korea; my97@honam.ac.kr; 2Department of Urban Policy Research, Goyang Research Institute, Goyang 10393, Korea; kej@goyang.re.kr; 3MECnWIP, Daejeon 35316, Korea; jssoyeon25@gmail.com; 4Global MICE Major, Dongduk Women’s University, Seoul 02748, Korea

**Keywords:** job satisfaction, job performance, professionalism perception, convention workers, cluster analysis

## Abstract

In recent years, the importance of professionalism of convention workers has been rapidly emerging. Therefore, the purpose of this study is to establish a strategy for strengthening the professionalism of convention workers. For this, the study investigates if there are any differences in job satisfaction and job performance based on the segmented groups of professionalism perception of convention workers. The results of factor analysis showed six underlying dimensions of professionalism perception of convention workers. Cluster analysis showed that there were different segmented groups of professionalism perception: high level (cluster 1), low level (cluster 2), moderate level (cluster 3). Lastly, MANOVA showed that there were differences in job satisfaction and job performance among the segmented groups. More theoretical and practical implications are discussed in the conclusion.

## 1. Introduction

The convention industry is a high value-added business since it improves the global image of a nation and creates additional value to various related industries. It is also considered as a major industry that creates economic and non-economic value [1,2]. As a global trend, the industry is strategically developed as a new growth engine industry along with the 4th industries, and the government is making an effort to establish institutional framework that may promote international competitiveness and increase economic impact through integration of MICE (Meetings, Incentives, Conventions, Exhibitions) infrastructure and convergence with other industries [3]. The regional MICE policies that should be prioritized are the expansion of infrastructures, such as convention and exhibition center and MICE facilities, and administrational support like establishing MICE organizations that attracts and aids MICE events and marketing the city as a MICE destination [4,5]. In Korea, the convention industry was chosen as the high added-value business among new growth engine industries, and the country is consolidating its position as one of the major convention hosting countries.

For the growth of the convention industry, expansion of hardware holds its importance, but software expansion is the one that should be weighed more. Interest in software ultimately promotes the competitiveness of the convention industry that is connected to voluntary participation of convention workers, which increases the possibility of management and attraction of international conventions [6]. This is because the convention companies are reliant upon human resources due to their nature as service business, and their success heavily depends on human resources and software. In other words, professionalism and competence of workers is the key to success and development of the convention industry.

Since the convention industry is a human-managed business and also a human-centered business, nurturing human resources who have received professional education is required, so that conventions can be systematically hosted and operated [7]. Compared to other hospitality industries, human resources are valued more, because every task is practically managed by workers [8]. For the development of the industry, workers should be educated as experts of the professional convention industry, which plays a central role in hosting and attracting conventions [9]. Convention industry workers may be called organizer, convention organizer, consultant, coordinator, convention host, convention designer, etc. [10]. They are experts that plan and provide a service related to convention hosting [11].

The workers undertake their job in areas of Convention, Exhibition, Incentive, and Meeting. Akgunduz, Kizilcalioglu, and Sanli, (2018) [12] define them as workers who plan and operate every activity related to convention and conference. From this, it may be inferred that there is an increase of perception that the convention workers should promote their professionalism, and the job itself has to be professionalized. From research related to job professionalism, the higher the professionalism is, the better the job satisfaction is and the deeper the workers are committed in their organization, which is likely to maximize the outcomes.

Professionalism is the attitude that encompasses dedication to the job, belief about the work, and the freedom of decision based on professional knowledge [13]. Therefore, offering a service without professionalism is limited to conveying a repeated and fragmented service. Hence, it is difficult to gratify the customer and job satisfaction, due to lack of pride and vocation as a convention worker [14]. Until now, in a knowledge-based society, a long-term and integrated nurturing plan that can create high value has been absent in Korea. Also, on account of limitations that educational institutes are facing, it is difficult to provide a reasonable and future-oriented education. In addition, the competitiveness of the industry has diminished due to the lack of education and training programs that enable a quick response to the rapidly changing social trends. Currently, efficient training of experts that may develop core competencies of the Korean convention industry is somehow hindered [15]. Also, compared to foreign countries, the research about effects of professionalism upon job satisfaction and organizational commitment is insufficient.

The originality of this study can be described as follows. Previous studies have used variables related to working conditions such as welfare and annual salary. However, in this study, the actual competency, in particular, was focused on the recognition of the professionalism required for the PCO (Professional Convention Organizer). The purpose of this study is to investigate the effects on job satisfaction and job performance according to the perception of professionalism and the analytical contents resulting from the differences. External issues such as working conditions are not easily resolved in the MICE industry. Therefore, we considered how to achieve performance through internal change and improvement targeting PCO with high turnover intention and high dissatisfaction.

The purpose of this study is to identify the professionalism perception of convention workers, and the difference between job satisfaction and job performance depending on the perception. Based on the results, we aim to conceptualize the professionalization perception of convention workers and put forward suggestions regarding content and direction for the ideal convention expert of today, and to present practical and academic implications related to human resource development of the convention industry.

## 2. Literature Review

### 2.1. Professionalism Perception of Convention Workers

Convention industry workers do their job in the areas of meeting, incentive, convention, and exhibition. There are various bodies that are encompassed in the industry. CVB (Convention and Visitors Bureau) hosts and manages conventions. Academic societies, institutes, governments, associations, organizations are the main bodies that host conventions and create demand. PCO and brokerages do the work for them. There are also owners of convention centers, and support companies. Convention workers may be defined as people who provide labor and earn a wage from a convention business [16]. The main job of a convention worker includes preliminary meetings, field activity, follow-up meetings, agenda planning, venue selection, budget establishment, and negotiation. It is classified into association convention planning and company convention planning in which the worker takes full responsibility [17]. It may also be called convention staff, convention manager, and convention executive. The associated role may be categorized as communication experts, operators and organizers, information experts, and management consultants [18]. Related occupations are PCO, PEO, workers of convention centers, event organizers, CVB, etc. [19]. Among various convention workers it can be said that the PCO play the most important role. The US Department of Labor and the Dictionary of Occupational Titles describes it as the job that organizes and plans every activity that realizes conventions and meetings [20]. There are opinions that in the convention industry, the human service provided by the workers is the first factor to be considered (i.e., more important than hardware). This is because the quality perception of consumers and integrated image management relies on the performance level of convention staff. Convention workers have diverse duties, and there are various abilities are required to be a convention project manager.

The lexical meaning of professionalism refers to professional traits, qualities, and the ability to outperform average people in specific areas, which may be acquired by systematic and long-term training [21]. The concept of professionalism was differently defined depending on scholars and the time period. Based on features and perspective about professionalism, it may be divided into three generations [22]. The first generation starts from the research which focused on problem-solving and information-processing abilities. It asserted that there are no differences between experts and non-experts, and experts refers to people who excel in general exploration and discovery, since they show superior performance in specific ability related to games. Holyoak (1991) [22] presented exceptional situations of the second-generation professionalism research and insisted the need for a third-generation research. The third-generation research defined experts as people solving specific problems, which refers to people who can meaningfully connect small units of information and knowledge. The second-generation studies conducted by Anderson (1982) [23], Rosenbloom and Newell (1987) [24], and Shiffrin and Schneider (1977) [25], revealed that experts may recognize and solve problems more precisely and faster compared to non-experts by utilizing exclusive knowledge. That is, the comparative study about the problem-solving ability of non-experts and experts dispatches professionalism into a unique area [22]. The components of professionals were divided into the structural side, organizational side, and attitudinal side and they were analyzed. In one of the widely used concepts, attitudes are classified into: utilization of organization, principle about self-regulation, autonomy, principle about public services, and vocation.

Professionalism perception is defined as “The perception about the exclusive authority, profession, and autonomy about the job, the pursuit of public interest, vocation, and the belief to self-control” [26] along with subconscious factors such as thoughts and opinions about the job of oneself as a professional. It defined professionalism as being qualified with an autonomous organizational system and professional knowledge earned through long-term training and taking responsibility [27]. Human resources is recognized as an important factor in the rapid growing Korean convention industry, but the professionalism of the workers and the index of that shows that social perception of it as experts is lacking. Therefore, it is time for studies related to human resources to expand their scope. Convention companies are making efforts to secure professionalism of workers through competency development, since they recognize the importance of the competency of PCO, because it determines the outcomes of companies achieved by successful hosting of conventions and secures competitiveness in the convention industry that is sensitive to fast changing market and has harsh competition [28]. Professionalism is categorized into different factors, depending on which part is emphasized [29]. Kang and Ritzhaupt (2015) [30], the early researchers of professionalism, said that three factors of systematic and professional knowledge, authority and privilege as a professional and social agreement, and social contributions of services provided by professionals should be considered. Since the components of professionalism are differently required depending on the periodical changes and characteristics of organizations, there are different academic opinions. Thus, based on preceding research, this study defines convention workers as people who plan, organize, operate, and evaluate every activity related to convention hosting. The professionalism perception of convention workers is divided into six subfactors: Vocational expertise, Work autonomy, Social contribution, Social status, Job ethics, and Work functionality, for verification. After subdividing the professionalism perception, its effects on Job satisfaction and Job performance may be analyzed.

### 2.2. Job Satisfaction

Job satisfaction is an important concept from a managerial viewpoint, and it is differently defined depending on the viewpoint of researchers. Gilmer (1996) [31], state that satisfaction or dissatisfaction is a result of personal factors related to the job and various attitudes regarding daily life. Loker (1976) [32] define it as a general attitude about the job and the emotional state that positively percepts the job experience.

Marhieu, Hofmann and Farr (1993) [33] define job satisfaction in three viewpoints. First, job satisfaction is a set of different attitudes toward job and job condition. Second, it may be decided by the comparison of expectations and actual experiences. Third, the satisfaction is decided by various factors such as wage, job, supervisors and coworkers, chances of promotion, etc. Heskett, Sasser and Schlesinger (1997) [34] considered that the increase of job satisfaction also naturally includes the loyalty of the workers, which naturally leads to improvement of the perceived quality of service provided to customers.

The increase of job satisfaction has positive effects in development of organizations, such as promoting mental health, work outcome, motivation, and quality of life. On the other hand, the decrease has negative effects like causing turnover intention, stress, etc. Also, it was discovered in preceding research [34] that the job satisfaction affects the quality of service. Based on the discovery, job satisfaction has to be gratified to realize an increase in outcomes.

Locker (1990) [35] identified the factors of job satisfaction as: wage, promotion, welfare, work condition, duty, supervisor, company, coworker, and managing method of the company. Poter and Steers (1983) [36] identified them as work condition factor, work content factor, and overall organization factor. Work condition factors are: participatory decision-making, the scale of working group, relation with coworkers, and type of supervision. Work content factors are: role conflict, working environment, ambiguity in role, and personal traits. Lastly, the overall organization factors are: wage, policy, procedure, and chances of promotion.

Among various factors that improve the outcome of the organizations when personal aims and values are gratified by organizational support are: high job satisfaction, organizational commitment, and active attitude towards organizational goals [37]. The degree of satisfaction is an important criterion for management and evaluation of organizations. In the worker’s point of view, important factors are: mental health, physical health, and value judgement [38]. In the organization’s point of view, workers that have positive attitude towards their job increase productivity by describing the organization positively and favorably and forming an amicable relation that decreases absence and turnover rate [39]. Therefore, it can be said that the higher the satisfaction is, the better the quality of service is. On this basis, there are claims that the managers can increase the quality of service by increasing the job satisfaction of their workers [40]. Kianto, Vanhala, and Heilmann (2016) [41] considered job satisfaction as a type of attitude, and defined it as the attitude of individuals toward their jobs. Mathieu, Fabi, Lacoursière, and Raymond (2016) [33] defined job satisfaction as the job of individuals or feelings and responses of individuals toward their jobs. In addition, it was defined as attitude of individuals toward factors related to their jobs such as wage, work, supervision, etc. [42]. Atmojo (2015) [43] said that job satisfaction means the emotional evaluation of the job that the worker holds. Bin (2015) [44] defined it as emotional and sentimental satisfaction acquired from evaluation of work condition and reputation.

This study defines job satisfaction as the emotional attitude of convention workers toward their convention work, and the positive emotion that is felt depending on the degree of gratification of job-related desires of the workers. Based on preceding studies, the job satisfaction of convention workers is composed of three factors: work-related factors (promotion and duty), boss and colleague related factors, and wage and welfare related factors.

### 2.3. Job Performance

Organizational outcome refers to productivity, quality, customer satisfaction, etc. Financial outcome includes gain by investment, return on asset, etc. Human resource management outcome is the results of attitude and behavior of workers, encompassing job satisfaction, turnover intention, organizational commitment, etc. [45,46].

Job Performance is defined by various studies. Lawler and Porter (1967) [36] defined it as results of activities that may be objectively measured, which contributes to the degree of achievement of organizations. Tett and Mayer (1993) [47] defined as the sound status of the work that an individual wants to realize within the organization and the degree of achievement of individuals.

In various studies related to job performance, most factors are correlated with job performance. Managers recognize that the productivity increases when workers are satisfied with their job and their morale is boosted, and the overall outcome of the organization increases as the achievement level of satisfied workers is higher, and this opinion is widely accepted [48].

This study defines the job performance of the organizers as the degree of achievement connected with individual jobs or achievement level of assigned tasks in which the results are self-recognized. Preceding studies show that various factors affect job performance. Managers thought that job satisfaction and high morale results in an increase of productivity. This is based on the assumption that the satisfied workers would work diligently and increase the outcomes of the organization, which was recognized as a universal proposition [49]. The measurement factors of job performance were analyzed under different criteria of researchers.

Organizational commitment was vastly researched in areas like sociology, industrial psychology, and behavioral science, and is considered to be one of the important indicators that show the outcomes of the organization [50]. In particular, much follow-up research has been conducted since it is more efficient than job satisfaction when predicting turnover intentions, acts as a useful indicator that shows the organizational effectiveness, and shows the relation between behavior and attitude of workers due to its mid and long-term stability [51]. Pelit, Dinçer, and Kılıç (2015) [52] defined it as psychological traits related to attitudes and behaviors of workers, such as the mindset to make efforts for the organization, loyalty, strong desire to remain in the organization, obsession, and identification of workers with organization. Ayazlar and Güzel (2014) [53] defined it as identification with organization and the relative intensity of commitment.

In addition, job performance was measured by factors of job satisfaction, customer service, organizational commitment, and achievement of goals. This study defined job performance as the degree of achievement related to convention work in which should be a self-recognized result. Based on preceding research, the job performance of convention workers is divided into 5 items: driving force in work, level of completion, level of achievement, timeliness, and sense of responsibility. The difference in job performance depending on the subdivision of professionalism perception of convention workers will be verified.

## 3. Methods

### 3.1. Research Questions

The purpose of this study is to identify job satisfaction and job performance based on the type of professionalism perception model of convention industry workers. In order to achieve such a purpose, the study was designed to answer the following research questions:Q1.Is the job satisfaction of convention workers different among groups of professionalism perception?Q2.Is the job performance of convention workers different among groups of professionalism perception?Q3.Are the demographics of convention workers different among groups of professionalism perception?

### 3.2. Generation of Questionnaire Items

The questionnaire items are organized based on the preceding research introduced above. In total, 24 of them are about professionalism perception of convention industry workers, 12 are about job satisfaction, and 5 are about job performance. The 5-point Likert scale is utilized for answers. Items about professionalism perception used preceding research [54,55] as reference. Items of preceding research about job satisfaction [56] and job performance [57] were modified and supplemented in a way that matches the purpose of this research and then inserted in the questionnaire. Lastly, demographic items that identify information of respondents such as gender, age, monthly income, job, and area of residence were added in order to understand the characteristics of the samples. The questionnaire items on professionalism perceptions identified 7 factors (vocational expertise, work autonomy, social contribution, social status, job ethics, work functionality) from Lee, S.S., Park, E.M. (2014); Lim, D.J., Ha, H.S., and Kim, H.Y. (2017) [55,56]. The job satisfaction questionnaire identified three factors (salary satisfaction, work contents satisfaction, relationship with coworkers’ satisfaction) from Lee, M.H., Chang, H.J. (2015) [57]. The job performance questionnaire identified one factor from Sohn, S.A. (2017) [58].

### 3.3. Data Collection and Analysis

The sample analyzed in the research has a size of 242 and was collected among convention industry workers. Researchers explained the purpose to the subjects, and upon their agreement it was conducted in a self-administered method. The research materials were collected from 4th October 2018 to 5th December 2018 both online and offline. Collected materials were analyzed with Statistical Package for the Social Sciences (SPSS) 21 Program. The collected data were analyzed by conducting frequency analysis to understand the characteristics of the data. To evaluate the validity and reliability, principal factor analysis with varimax rotation was conducted on all statements. The hierarchical cluster analysis using the Ward method and the K-means cluster analysis were combined to subdivide the group according to the job characteristics of the convention project manager. Discriminant analysis was then applied to detect whether any significant differences existed among the three groups. Cross-tabulations analysis was performed to verify the difference in demographic profile according to the job characteristics of the convention project manager. Finally, multivariate analysis of variance (MANOVA) and analysis of variance (ANOVA) were conducted to verify the difference in core competencies for each group according to the recognition of job characteristics of convention workers.

## 4. Results

### 4.1. Profile of Respondents

Demographic identification results show that a slight predominance of males, since it is shown that 54.2% of the respondents were male and 45.8% were female. In total, 32.0% of them were in their 30s, which was the most common age group, followed by the order of 20s, 40s, and 50s. Most of them were university graduates (64.5%) or graduate school graduates (17.4%), so it can be said that they have a high level of education. A majority of them earned 2 million KRW (Korean Won), ~less than 3 million KRW (37.6%) or less than 2 million KRW (33.1%) as monthly income. Lastly, working years is aligned in order of 2~less than 4 years (26%), 4~less than 6 years (21.5%), less than 2 years (19.4%), over 10 years (10.3%), and 8~less than 10 years (9.1%). For their position, a large number of respondents were deputy (26%), manager (23.1%), and director or higher (13.6%) (Table 1).

### 4.2. Verification of Reliability and Validity of Fprofessionalism Perception and Job Performance of Convention Industry Workers

In order to verify the reliability and validity of indexes used to measure the construct of professionalism conception of convention industry workers, exploratory factor analysis and reliability analysis were used (Table 2).

Exploratory factor analysis was initiated utilizing 24 items of professionalism perception. Under the Varimax rotation method, factors with an Eigen-value of 1 or higher were extracted. As a result, variables with a factor loading value of 0.4 or lower were excluded. Thus, among 20 variables, 6 factors were deduced, and they were named ‘Vocational expertise’, ‘Work autonomy’, ‘Social contribution’, ‘Social status’, ‘Job ethics’, and ‘Work functionality’. The total variance explanation power is 70.17%, and the results of KMO (Kaiser-Meyer-Olkin) measure of sampling adequacy appears as a value of 0.798, Bartlett’s test sphericity demonstrates a significance at a level of 0.000 (χ^2^ = 2592.661, df = 190). In order to examine the internal consistency, the Cronbach’s α results were reviewed, and all of them showed 0.6 or higher, which shows that it is reliable [59].

As a result of the verification of reliability and validity of job performance of convention industry workers (Table 3), 3 factors were deduced and were named ‘salary satisfaction’, ‘work contents satisfaction’, and ‘relationship with coworkers’ satisfaction’. The total variance explanation power was 73.44%, and the results of KMO measure of sampling adequacy generated a value of 0.806 and Bartlett’s test sphericity demonstrates a significance at a level of 0.000 (χ^2^ = 1114.210, df = 45). In order to examine the internal consistency, the Cronbach’s α were viewed and all of them showed 0.6 or higher, which proves its reliability.

### 4.3. Cluster Analysis Based on the Professionalism Perception of the Converntion Workers

In order to obtain a subdivided cluster for the professionalism perception of convention industry workers, a cluster analysis was initiated utilizing relevant variables (Table 4). To determine the optimal number of clusters, Ward’s hierarchical cluster analysis method was used simultaneously with the K-means non-hierarchical cluster analysis method. The 6 factors for professionalism perception of convention industry workers were hierarchically analyzed and categorized into 3 clusters by referring to fall range and rate that shows the variance of dendrogram and coefficient. Then it was designated into 3 clusters and analyzed by utilizing the K-Means method.

The results of MANOVA show that the different groups had significant differences in their awareness toward the professionalism (Table 5). Cluster 1 (*n* = 111) is the high-level professionalism perception group. Cluster 2 (*n* = 61) is the low-level professionalism perception group and Cluster 3 (*n* = 70) is the moderate-level professionalism perception group.

To approve the viability of the categorized clusters, a discriminant analysis was initiated (Table 6). In the analysis sample (*n* = 242) of Table 6, which is used to derive the discriminant function, 91.0% of the cases of Cluster 1, specifically 101 out of 111, were correctly allocated. In Cluster 2 93.4% of the cases, exactly 57 out of 61 cases, and in Cluster 3 was 69 out of 70, which is 98.6%, were correctly allocated. It can be said that it is highly precise since a hit ratio of 93.8% is shown among the 242 respondents.

### 4.4. Difference of Job Satisfaction and Job Performance by Cluster

The MANOVA results of job satisfaction of convention industry workers of different professionalism perception clusters is as follows (Table 7). Job satisfaction was classified into 3 factors: salary satisfaction, work contents satisfaction, and relationship with coworkers’ satisfaction. As a result, the Wilks’ lambda = 0.631 (F = 20.447, *p* < 0.001) shows that the job satisfaction factors were found to be significantly different among professionalism perception clusters. Salary satisfaction and Work contents satisfaction were generally higher in cluster with high professionalism perception. However, Relationship with coworkers’ satisfaction was the highest in the cluster with moderate professional perception. It can be interpreted from this that the satisfaction that comes from relationships with coworkers is related to the job satisfaction, even if the professional perception is not at a high level.

The MANOVA result of job performances of convention industry workers of different professionalism perception clusters is as follows (Table 8). As a result, F = 35.026, *p* < 0.001. The job performance factors were found to be significantly different among professionalism perception clusters. It can be confirmed that the higher the professionalism perception of convention industry workers is, the higher the job performance is.

### 4.5. Cross Tabulation of Professionalism Perception Segmentation and Demographic Characteristics

In order to initiate an in-depth analysis of 3 categorized demographic characteristics of different professionalism perceptions, a cross tabulation was conducted. Education and Position were utilized as variables to analyze characteristics of clusters, and there was a statistically significant characteristic of clusters in education. In general, university graduation plays a substantial part in the 3 clusters. However, there are differences in other education. Cluster 1 had the highest portion of postgraduates or above (21.6%) and Cluster 2 dominated in terms of university graduation. Cluster 3 had both 17.1% of college graduation and postgraduate or above, which is relatively higher compared to the others (Table 9).

## 5. Discussion and Implications

### 5.1. Discussion of Results

There are three implications of this study. First, there have been subdivisions in professional perception of convention workers and the difference by group was recognized. There are six factors regarding the professional perception that was established, namely: Vocational expertise, Work autonomy, Social contribution, Social status, Job ethics, and Work functionality. This supports the results of the study [26]. Through cluster analysis and discriminant analysis, three clusters (clusters with high level of professionalism perception, moderate level of perception, and low level of perception) were deduced and through verification tools, the difference in perceptions was observed. Second, the difference of job satisfaction and job performance depending on professionalism perception by clusters was verified.

The result shows that the cluster with a high level showed high salary satisfaction and had the highest work contents satisfaction. This somehow shows that the high level of perception is matched with appropriate wage level and satisfaction in daily life. In addition, they have the feeling that their job is valuable and are interested by their work, which is followed by sense of fulfillment. These results support the research results and claims of previous studies [37,49] presented above.

However, the high level of perception did not show a relationship with coworkers’ satisfaction, and the cluster with low perception showed a low level of job satisfaction in every aspect. Third, the difference of clusters by demographic characteristics was verified. In case of monthly wage, the perception level decreased as the monthly wage increased. This shows the characteristic of the MICE industry. The increase in monthly wages does not secure professionalism, but instead active work experience somehow accumulates professionalism. In addition, it reflects the increasing turnover intention caused by the low increase rate of wages. This result was found to support some of the findings of the literature review [14].

There may be five implications presented about the job satisfaction and job performance difference depending on professionalism perception of convention workers. First, there should be motivation and education that could promote professionalism perception towards clusters with low perception. This may improve job satisfaction and job performance. Second, since old age and high numbers of service do not prove high professionalism perception, promotion in appropriate period or provision of environment that ensures both work and study is required. Third, to increase the professionalism perception of convention workers, improvements of working conditions and institutional supplementation of wage, welfare, and job security compared to other occupations is needed. Fourth, in order to promote professionalism of convention workers, their intellectual curiosity should be gratified by offering the chance of participation in seminars, annual general meetings, forum and communities that are hosted by governance level institutions like government and associations. Fifth, work standardization indexes should be supplemented to increase professionalism of convention workers, and work and career development of the industry should be systemized by encouraging companies to utilize work standardization and behavioral standards.

### 5.2. Limitations and Future Research

Through this study, academic and practical significance was found in human resource management in the MICE industry, but several limitations exist. Therefore, a follow-up study that can complement and develop these limitations is proposed. First, in this study, a quantitative survey method using a questionnaire was performed to find out the difference between job satisfaction and job performance according to professional recognition. By expanding its scope nationwide, we can understand the perception and underlying psychology of the value of MICE work, identify regional disparities, and discuss solutions. In addition, it is expected that results that can facilitate understanding, and the development of various occupations in the convention industry can be derived through research on the differences between various occupational groups in the MICE industry.

## Figures and Tables

**Table 1 ijerph-19-05829-t001:** Profile of respondents.

Item		Frequency	Ratio	Item		Frequency	Ratio
Sex	Male	131	54.1	Age	20s	98	40.5
	Female	111	45.9		30s	96	39.7
Monthly income	less than 2 million W	80	33.1		40s	38	15.7
	2 million W~less than 3 million W	91	37.6		Over 50	10	4.1
	3 million W~less than 4 million W	42	17.4	Position	Employee	57	23.6
	4 million W~less than 5 million W	18	7.4		Chief	33	13.6
	Over 5 million W	11	4.5		Deputy	63	26
Working years	less than 2 years	47	19.4		Manager	56	23.1
	2~less than 4 years	63	26.0		Director or higher	33	13.6
	4~less than 6 years	52	21.5	Education	high school diploma or under	16	6.6
	6~less than 8 years	33	13.6		college graduation	28	11.6
	8~less than 10 years	22	9.1		University graduation	156	64.5
	Over 10 years	25	10.3		Postgraduate or above	42	17.4

**Table 2 ijerph-19-05829-t002:** Results of factor analysis for Professionalism perception variables.

Factor	Mean	Factor Loading	Eigen Values	Explained Variance	Cronbach’s α
Factor 1: Vocational expertise			6.029	30.147	0.830
Participation in outside activities is needed to improve professionalism	4.17	0.808			
Qualification standards for job related certificates have to be strengthened to improve professionalism	4.14	0.798			
Dedicated organization is needed to improve welfare and professionalism	4.24	0.770			
Academic degree related to the job is required	3.96	0.726			
Factor 2: Work autonomy			2.421	12.105	0.894
The importance of tasks of convention workers is acknowledged	4.05	0.872			
Convention management is left to the convention workers	4.07	0.866			
Convention workers take responsibility for incidents that have occurred during the convention	4.02	0.712			
Factor 3: Social contribution			1.764	8.819	0.802
The job contributes to the cultural development of the local society	4.32	0.779			
The job contributes to the economic development of the local society	4.31	0.748			
The job contributes to development of future society	4.20	0.712			
The job requires social contribution over economic gains	3.98	0.630			
Factor 4: Social status			1.502	7.511	0.800
Working condition is fine	2.54	0.901			
Have better incomes compared to other occupations of similar academic background	2.60	0.823			
Job security of the industry is stable	2.59	0.794			
Factor 5: Job ethics			1.201	6.004	0.705
Subcontractors are respected, and their differences are accepted	4.26	0.841			
There are beliefs in work that mediate and manage affairs like convention hosting, organizing, operating, etc.	4.26	0.769			
Moral obligations are well followed	4.21	0.580			
Factor 6: Work functionality			1.117	5.584	0.625
Voluntary participation in various educations is required	4.20	0.797			
High-level of work performance is required	4.18	0.703			
Continuous research about method of mediation and management of works such as international conference hosting, organizing, operating is needed.	4.24	0.659			

Note: KMO measure of sampling adequacy = 0.798, Chi-Square = 2592.661, df = 190, *p* < 0.001.

**Table 3 ijerph-19-05829-t003:** Results of factor analysis for job performance variables.

Factor	Mean	Factor Loading	Eigen Values	Explained Variance	Cronbach’s α
Factor 1: salary satisfaction					
Satisfied with the decision procedure of wage, bonus, and allowance	2.70	0.895	3.443	34.430	0.890
The effort matches the wage	2.71	0.875			
Wage is enough for living	2.78	0.866			
Satisfied with welfare	2.81	0.832			
Factor 2: work contents satisfaction					
The job is valuable	4.33	0.862	2.757	27.566	0.831
The job provokes constant interest	4.28	0.829			
Feel sense of accomplishment from the job	4.43	0.821			
Factor 3: relationship with coworkers’ satisfaction					
Coworkers are helpful	4.16	0.808	1.144	11.445	0.752
My boss evaluates me fairly	3.77	0.792			
Satisfied with relation with coworkers	4.04	0.778			

Note: KMO measure of sampling adequacy = 0.806, Chi-Square = 1114.210, df = 45, *p* < 0.001; Bartlett’s test of sphericity (*p* < 0.001).

**Table 4 ijerph-19-05829-t004:** Results of Cluster analysis based on Professionalism perception.

		Vocational Expertise	Work Autonomy	Social Contribution	Social Status	Job Ethics	Work Functionality
K-means cluster analysis	cluster 1 (*n* = 111)	4.33	4.52	4.48	3.11	4.34	4.24
	cluster 2 (*n* = 61)	3.59	2.84	3.52	2.40	3.88	3.98
	cluster 3 (*n* = 70)	4.28	4.35	4.35	1.87	4.41	4.35
	F	42.030 ***	238.378 ***	73.285 ***	84.911 ***	21.858	9.763 ***

Note: *** *p* < 0.001, cluster 1: High level, cluster 2: Low level, cluster 3: Moderate level.

**Table 5 ijerph-19-05829-t005:** Results of differences between clusters by Professionalism perception.

Function	Cluster	Mean	S.D.	F	*p*	Post-Hoc Analysis
Vocational expertise	1	4.327	0.050	42.030	0.000	1 < 3 < 1
	2	3.590	0.068			
	3	4.282	0.063			
Work autonomy	1	4.523	0.048	238.378	0.000	2 < 3 < 1
	2	2.836	0.065			
	3	4.352	0.060			
Social contribution	1	4.482	0.048	73.285	0.000	2 < 3 < 1
	2	3.525	0.065			
	3	4.346	0.061			
Social status	1	3.111	0.060	84.911	0.000	3 < 2 < 1
	2	2.404	0.081			
	3	1.871	0.076			
Job ethics	1	4.342	0.048	21.858	0.000	2 < 1 < 3
	2	3.880	0.064			
	3	4.410	0.060			
Work functionality	1	4.243	0.047	9.763	0.000	2 < 1 < 3
	2	3.978	0.063			
	3	4.348	0.059			

Note: Wilks’ Lambda = 0.166, F = 56.85, *p* < 0.001.

**Table 6 ijerph-19-05829-t006:** Results of Discriminant analysis for clusters.

Function	Eigen Value	Dispersion (%)	Canonical Correlation	Wilks Lambda	df	χ^2^
1	2.584	80.2	0.849	0.170	8	420.541 ***
2	0.639	19.8	0.624	0.610	3	117.381 ***
cluster title		cluster 1 (*n* = 111)	cluster 2 (*n* = 61)	cluster 3 (*n* = 70)		Total
High-level		101 (91.0%)	3 (4.9%)	1 (1.4%)		111
Low-level		1 (0.9%)	57 (93.4%)	0 (0.0%)		61
Moderate-level		9 (8.1%)	1 (1.6%)	69 (98.6%)		70

Note: *** *p* < 0.001, Discriminatory Hit ratio 93.8%.

**Table 7 ijerph-19-05829-t007:** Results of differences of job satisfaction by cluster.

Function	Cluster	Mean	S.D.	F	*p*	Post-Hoc Analysis
Salary satisfaction	1	4.52	0.44	28.13	0.000 ***	2 < 3 < 1
	2	3.93	0.64			
	3	4.42	0.47			
Work contents satisfaction	1	3.11	0.80	28.39	0.000 ***	3 < 2 < 1
	2	2.72	0.84			
	3	2.21	0.71			
Satisfaction with Relationship with coworkers	1	4.09	0.56	15.84	0.000 ***	2 < 1 < 3
	2	3.62	0.64			
	3	4.16	0.64			

Note: *** *p* < 0.001; Wilks’ Lambda = 0.631, F = 20.447, *p* < 0.001.

**Table 8 ijerph-19-05829-t008:** Results of differences of job performance by cluster.

Function	Cluster	Mean	S.D.	F	*p*	Post-Hoc Analysis
Job performance	1	4.42	0.44	35.026	0.000 ***	2 < 3 < 1
	2	3.82	0.51			
	3	4.33	0.47			

Note: *** *p* < 0.001.

**Table 9 ijerph-19-05829-t009:** Results of Cross Tabulation of Professionalism Perception Segmentation and Demographic Characteristics.

Characteristic		Professionalism Perception of the Convention Workers			χ^2^	*p*
		Cluster 1	Cluster 2	Cluster 3		
Education	High school graduation	7 (6.3%)	1 (1.6%)	8 (11.4%)	17.003	0.009 **
	College graduation	13 (11.7%)	3 (4.9%)	12 (17.1%)		
	University graduation	67 (60.4%)	51 (83.6%)	38 (54.3%)		
	Postgraduate or above	24 (21.6%)	6 (9.8%)	12 (17.1%)		
Position	Employee	24 (21.6%)	23 (37.7%)	10 (14.3%)	15.188	0.056
	Chief	15 (13.5%)	8 (13.1%)	10 (14.3%)		
	Deputy	27 (24.3%)	16 (26.2%)	20 (28.6%)		
	Manager	32 (28.8%)	7 (11.5%)	17 (24.3%)		
	Director or higher	13 (11.7%)	7 (11.5%)	13 (18.6%)		

Note: ** *p* < 0.01.

## Data Availability

Not applicable.

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
