# Peer review of "Toward Improving Working Conditions to Enhance Professionalism of Convention Workers: Focusing on the Difference between Job Satisfaction and Job Performance According to Professionalism Perception"

_ijerph, 2022, doi:10.3390/ijerph19105829_

Round 1

Reviewer 1 Report

The research done is interesting. It is well structured, the methodology used to meet the objective is adequate and the review of the literature is sufficient.
In order for authors to carry out the process of improving the document below, I suggest the following recommendations:
one/. The abstract should be rewritten, in which the research should be presented in a clear and concise manner. First of all, the objective is not clearly stated, it is more clearly described in the introduction section. Therefore, the objective should be clearly stated in both the abstract and the introduction section. The methodology used should also be clearly stated in a short and concise manner. Present the results obtained in the research clearly and not linked as they are to the methodology. Scheme: Contextualization, Objective, methodology and results.
two/. The introduction section needs to be rewritten, relevant information is missing in the current state.
- The problem under study must be adequately contextualized.
- It is necessary to identify based on which research a gap in the literature is identified and therefore the authors propose to carry out the study. What studies have been carried out so far on the problem to be studied? Why is it necessary to carry out the proposed research? What is the gap identified in the literature?
- Indicate clearly, what is the novelty of the investigation?
- Include a final paragraph describing the sections into which the document is divided.
3./ The section of conclusions is a mere repetition of the results obtained. It is recommended to include a new section "Discussion" when comparing the results obtained with those of other studies to see if they are corroborated or not. Explain the same.
Include the Conclusions section: in not repeating the results again but rather presenting the conclusions obtained in a general way. It is very important that the Conclusions, Theoretical Implications and Management Implications appear clearly.
Put the limitations section at the end of the document, after the conclusions.

I wish the authors encouragement to improve the document so that it acquires the level of quality necessary for their document to be published in this prestigious journal.

Author Response

  1. one/. The abstract should be rewritten, in which the research should be presented in a clear and concise manner. First of all, the objective is not clearly stated, it is more clearly described in the introduction section. Therefore, the objective should be clearly stated in both the abstract and the introduction section. The methodology used should also be clearly stated in a short and concise manner. Present the results obtained in the research clearly and not linked as they are to the methodology. Scheme: Contextualization, Objective, methodology and results.

-> In the abstract, the purpose of the study was clarified and rewritten concisely.
The purpose of the study was clarified in the introduction, and a concise and clear explanation of the methodology was added.

2. two/. The introduction section needs to be rewritten, relevant information is missing in the current state.
- The problem under study must be adequately contextualized.
- It is necessary to identify based on which research a gap in the literature is identified and therefore the authors propose to carry out the study. What studies have been carried out so far on the problem to be studied? Why is it necessary to carry out the proposed research? What is the gap identified in the literature?
- Indicate clearly, what is the novelty of the investigation?
- Include a final paragraph describing the sections into which the document is divided.

-> The distinction from previous studies and the purpose of the study were clearly stated.

3. 3./ The section of conclusions is a mere repetition of the results obtained. It is recommended to include a new section "Discussion" when comparing the results obtained with those of other studies to see if they are corroborated or not. Explain the same.
Include the Conclusions section: in not repeating the results again but rather presenting the conclusions obtained in a general way. It is very important that the Conclusions, Theoretical Implications and Management Implications appear clearly.
Put the limitations section at the end of the document, after the conclusions.

-> The discussion section and the section on the limitations of the study were divided and rewritten.

Author Response

  1. introduction

-> The purpose of the study was clarified in the introduction, and a concise and clear explanation of the methodology was added.

2. methods

-> In order to prove the validity of the measured variables, references from which the measured variables were derived are indicated. A description of the analysis method used in this study has been added.

3. conclusions

->The discussion content and limitations of the study were additionally written.

Reviewer 3 Report

Thank you for the opportunity to review this work.
I propose to make the following changes:
1. The title of the work is too long and complicated - I propose to shorten it
2. In the introduction, expand the acronym MICE (Meetings, Incentives, Conferences & Exhibitions). In subsection 2.1, instead of the MICE acronym, an expansion is used. Therefore, it is difficult to understand why the acronym is introduced earlier and then it is not used. The acronym PCO, KRW, KMO is also used without its direct explanation, such as the Convention and Visitors Bureau (CVB). Please believe that not everyone knows these shortcuts.
3. There is a quote in lines 130-133. Whose quote is this? Does the given footnote [26] refer to the quotation or the opinion from the second part of the sentence? Why do the article use two citation systems? Sometimes these are the numbers to be quoted, and other times the names of the authors with the year of publication.
4. What is the reliability of the research carried out for the sample of 242 employees? What is the trial error? The limit value of the Cronbah alpha coefficient is 0.7, why is the level of 0.6 already considered reliable? (line 314).
5. The authors have forgotten about the Discussion chapter. Since this is an article, a discussion of the results in the context of the cited literature should be inserted (after Results).
6. Perhaps it will also be useful to take into account other surveys on employee satisfaction with work

Author Response

  1. title

-> The title has been edited for simplicity and clarity.

2. introduction

-> The abbreviation has been clarified so that the reader can understand it well.

3. quote

-> This part is a part of the literature search on the definition of professionalism, and it has been modified to clearly indicate it because there is a lot of content and it may confuse the reader.

4. reliability

-> In reliability analysis, the Cronbach alpha coefficient is judged to be acceptable if it is 0.6 or more and less than 0.7, good if it is 0.7 or more and less than 0.8, and excellent if it is 0.8 or more and less than 0.9. [59] Kim, D.M(2019), QuickPass Paper Statistical Analysis, https://qpassthesis.tistory.com/ 

5. Discussion chapter

-> The section and the limitations of the study were additionally written.

Round 2

Reviewer 1 Report

All comments and suggestions have been made correctly.

Author Response

Thanks for the comments. For some results, we add content supporting the previous research and resubmit it. Again, thank you very much for the good point.

Reviewer 2 Report

I consider the review carried out to be adequate. The methodology has been improved.

Author Response

(The authors gave the same response as above.)

Reviewer 3 Report

Thank you for the improvement, although I am surprised by the minimalism of the authors' actions, both in terms of the answers and the comments made.
The discussion is an exchange of views, so the Discussion chapter should contain references to what the authors cited in the literature review. Please complete the Discussion chapter with such references to the previously cited literature items.

Author Response

I deeply sympathize with the part you pointed out, and I wrote and resubmitted the part additionally. The reliability of the study could be further enhanced by revealing which previous studies could be supported for the results derived from this study. Once again, I would like to thank you very much for your good point. Thank you.